# Association of Early Childhood Caries with Bitter Taste Receptors: A Meta-Analysis of Genome-Wide Association Studies and Transcriptome-Wide Association Study

**DOI:** 10.3390/genes14010059

**Published:** 2022-12-24

**Authors:** Ekaterina Orlova, Tom Dudding, Jonathan M. Chernus, Rasha N. Alotaibi, Simon Haworth, Richard J. Crout, Myoung Keun Lee, Nandita Mukhopadhyay, Eleanor Feingold, Steven M. Levy, Daniel W. McNeil, Betsy Foxman, Robert J. Weyant, Nicholas J. Timpson, Mary L. Marazita, John R. Shaffer

**Affiliations:** 1Department of Human Genetics, University of Pittsburgh, Pittsburgh, PA 15260, USA; 2Bristol Dental School, University of Bristol, Bristol BS1 2LY, UK; 3Medical Research Council Integrative Epidemiology Unit, Department of Population Health Sciences, University of Bristol, Bristol BS8 1QU, UK; 4Dental Health Department, College of Applied Medical Sciences, King Saud University, Riyadh 12372, Saudi Arabia; 5Department of Periodontics, School of Dentistry, West Virginia University, Morgantown, WV 26505, USA; 6Center for Craniofacial and Dental Genetics, Department of Oral and Craniofacial Sciences, University of Pittsburgh, Pittsburgh, PA 15261, USA; 7Department of Preventive & Community Dentistry, University of Iowa College of Dentistry, Iowa City, IA 52242, USA; 8Department of Psychology & Department of Dental Public Health and Professional Practice, West Virginia University, Morgantown, WV 26505, USA; 9Center for Molecular and Clinical Epidemiology of Infectious Diseases, Department of Epidemiology, School of Public Health, University of Michigan, Ann Arbor, MI 48109, USA; 10Dental Public Health, School of Dental Medicine, University of Pittsburgh, Pittsburgh, PA 15213, USA; 11Avon Longitudinal Study of Parents and Children, University of Bristol, Bristol BS8 1QU, UK

**Keywords:** genetics, oral health, child, molecular epidemiology

## Abstract

Although genetics affects early childhood caries (ECC) risk, few studies have focused on finding its specific genetic determinants. Here, we performed genome-wide association studies (GWAS) in five cohorts of children (aged up to 5 years, total *N* = 2974, cohorts: Center for Oral Health Research in Appalachia cohorts one and two [COHRA1, COHRA2], Iowa Fluoride Study, Iowa Head Start, Avon Longitudinal Study of Parents and Children [ALSPAC]) aiming to identify genes with potential roles in ECC biology. We meta-analyzed the GWASs testing ~3.9 million genetic variants and found suggestive evidence for association at genetic regions previously associated with caries in primary and permanent dentition, including the β-defensin anti-microbial proteins. We then integrated the meta-analysis results with gene expression data in a transcriptome-wide association study (TWAS). This approach identified four genes whose genetically predicted expression was associated with ECC (*p*-values < 3.09 × 10^−6^; *CDH17*, *TAS2R43*, *SMIM10L1*, *TAS2R14*). Some of the strongest associations were with genes encoding members of the bitter taste receptor family (TAS2R); other members of this family have previously been associated with caries. Of note, we identified the receptor encoded by *TAS2R14*, which stimulates innate immunity and anti-microbial defense in response to molecules released by the cariogenic bacteria, *Streptococcus mutans* and *Staphylococcus aureus*. These findings provide insight into ECC genetic architecture, underscore the importance of host-microbial interaction in caries risk, and identify novel risk genes.

## 1. Introduction

Some of the most vulnerable members of society are impacted by a severe form of tooth decay termed early childhood caries (ECC). Defined as the presence of one or more decayed, missing, or filled tooth surfaces in any primary tooth in a child under six, this largely preventable disease affects more than 38% of children in the United States [1]. ECC is associated with multiple negative outcomes: pain, difficulties with chewing, self-esteem and behavior, sleep problems, and decreased school performance [2]. The harms are not limited to children but reverberate through the family. ECC causes emotional and financial stress for parents, and accessing dental care for children results in loss of workdays [2].

The most proximate cause of caries is the interaction of sugar intake and dysbiosis of the oral microbiome [3], but other factors influence susceptibility: genetics, other dietary content/patterns, toothbrushing, fluoride, and access to care. Caries is partially heritable, and displays higher heritability estimates for primary tooth caries than for permanent tooth caries [4]. Estimates for single nucleotide polymorphism (SNP)-based ECC heritability range from 13 to 53%, in line with reported SNP-based heritability estimates for primary tooth caries [5,6,7].

Genome-wide studies have successfully identified several genetic variants associated with caries. Although several genome-wide association studies (GWASs) of caries traits have been performed in children [5,8,9,10,11,12] and adults [6,13], the identified variants explain only a portion of the heritability, and few have been replicated, potentially owing to differences in the genetic architecture of caries among populations of adults and children. Furthermore, few studies have specifically focused on ECC and young children, in spite of the fact that earlier onset disease can display a higher genetic risk burden [14]. One previous pilot GWAS of ECC (*n*=212) was underpowered and as expected, did not find significant genetic associations [7].

Here, we performed GWASs of ECC in five cohorts, followed by a meta-analysis totaling 2974 children from a variety of socioeconomic backgrounds and geographic locations. We integrated these results with gene expression datasets to identify gene transcript-ECC associations.

## 2. Materials and Methods

### 2.1. Participant Cohorts

Research participants included in the study came from cross-sectional studies nested in five cohorts recruited from Pennsylvania, West Virginia, Iowa, and South West England. For additional details on cohorts and data collection please see the Appendix A.

The first cohort of the Center for Oral Health Research in Appalachia (COHRA1) is a cross-sectional study comprising members of households living in rural West Virginia and Western Pennsylvania—a region with a high prevalence of poor oral health outcomes [4,15]. It was designed to study the contributions of individual, family, and community factors to oral diseases. The dental examinations were performed by trained and calibrated dentists or dental hygienists during 2003–2009 for children aged 1–5. The coronal tooth surfaces were assessed, and decay was classified using the four-level classification method developed by the World Health Organization, including capture of both cavitated and white spot lesions. For children aged 1–3, a shortened lift-the-lip exam was performed instead of the protocol to document early childhood caries and tooth loss.

The second cohort of the Center for Oral Health Research in Appalachia (COHRA2) is a longitudinal study of the genetic, microbial, and environmental factors impacting oral health and is separate from COHRA1. Beginning in 2011, COHRA2 recruited women during pregnancy, and upon birth, their children, and followed them through age 6 of the child. Participants were ascertained in West Virginia and southwestern Pennsylvania. The cohort has been described previously [16]. Clinical dental examinations of babies were performed a month after the eruption of the first tooth (in the Pennsylvanian subset only), at one year, and every year thereafter (in all participants, Pennsylvanian and West Virginian). For this study, children’s caries data were used from the latest available visit prior to age 6 The caries exam was performed according to the PhenX Toolkit Dental Caries Experience Prevalence Protocol [17] (http://www.phenxtoolkit.org/, protocol number 080300, (accessed on 24 January 2011)), and simplified to be acceptable for dental evaluation in two-year-old children.

The Iowa Fluoride Study (IFS) was designed to study the relative importance of different fluoride exposures and intakes from dietary and non-dietary sources, and their relationships to dental fluorosis and caries [18,19]. It is a prospective cohort of mother–baby pairs recruited from eight Iowa hospitals between 1992 and 1995. The mothers were well educated [20] and of middle and high socioeconomic status. A trained and calibrated dentist performed a caries exam on each child aged 4–6 using portable equipment [18]. Criteria for assessing caries was based on published literature [18] and compatible with COHRA1 (and later, COHRA2).

The Iowa Head Start (IHS) study enrolled children via Iowa’s Heat Start, a federally funded program for low-income children. Children aged 3–5 were examined for caries during dental exams, and their DNA was collected using buccal brushes [21]. Dental exams were performed using a flashlight and mouth mirror [21]. Cavitated lesions were identified based on the combined d_2_-d_3_ criteria [22].

The Avon Longitudinal Study of Parents and Children (ALSPAC) is a prospective observational study of influences on health and development over the life course, which began recruiting pregnant women and eventually their children in 1990 in urban and rural areas of South West England [23,24] Dental exams were completed for a random 10% of the ALSPAC cohort by trained assessors at 31, 43, and 61 months of age. Related individuals were removed during quality control. All studies were approved by site- or university-specific institutional review boards, and parents in all studies provided informed consent. In ALSPAC, consent for biological samples was collected in accordance with the Human Tissue Act (2004). Informed consent for the use of data collected via questionnaires and clinics was obtained from participants following the recommendations of the ALSPAC Ethics and Law Committee at the time. The ALSPAC study website contains details of data that are available through a fully searchable data dictionary and variable search tool (http://www.bristol.ac.uk/alspac/researchers/our-data/ (accessed on 19 February 2018)).

### 2.2. Caries Phenotypes

ECC was defined according to the American Academy of Pediatric Dentistry as the presence of one or more decayed (white spot or cavitated lesion), missing (due to caries), or filled tooth surfaces (dmfs) in any primary tooth before age six years. The COHRA1 and COHRA2 cohorts classified both white spots and carious lesions as “decay,” whereas the IHS and ALSPAC cohorts only classified carious lesions and not white spots as “decay,” and the IHS and IFS cohorts did not have “missing due to caries” information. Children unaffected with ECC were used as the comparison group. Where multiple exams were performed, phenotypes were drawn from the last possible dental exam before age 6 (COHRA1, COHRA2, IFS).

### 2.3. Genome-Wide Association Studies (GWASs) and Meta-Analysis

Genome-wide association studies (GWASs) were performed in each cohort to identify common genetic variants associated with ECC. Genetic data for all cohorts were collected using whole-genome genotyping platforms and, to increase power of the GWAS analyses, imputed to 1000 Genomes Phase I (COHRA1, IFS, IHS) or the Haplotype Reference Consortium version 1.1 (COHRA2, ALSPAC) reference panels. See the Appendix A for details on dataset filtering, including standard filtering for common variants (minor allele frequency (MAF) > 5%), deviation from Hardy–Weinberg equilibrium, SNP and sample missingness, and imputation quality with thresholds specific to each cohort.

To minimize spurious genetic associations arising from confounding due to differences in allele frequency and trait distributions among individuals of different ancestries (termed population stratification), GWASs of ECC were limited to individuals of European ancestry (ALSPAC) or European non-Hispanic ancestry verified using principal component analysis relative to HapMap controls (COHRA1, COHRA2, IFS, IHS).

In COHRA1 and COHRA2, GWASs were run using linear mixed models adjusting for age, sex, and study site in Efficient Mixed-Model Association eXpedited (EMMAX) [25]. Linear mixed models account for population stratification and thus, do not require control for principal components of ancestry. In IFS and IHS, GWASs were run using logistic regression adjusting for age, sex, and one principal component of ancestry using PLINK v1.9 [26]. *p*-values were determined using adaptive permutation with a maximum of 1 M permutations per SNP to safeguard against false positive results. The ALSPAC GWAS was performed using linear mixed models adjusting for age and sex using BOLT-LMM for chromosomes one through twenty-two and using fast genome-wide association (fastGWA) in genome-wide complex trait analysis (GCTA) for the X chromosome (separately for females and males). Results for the X chromosome in females and males were combined via meta-analysis in PLINK v1.9. GWAS methods are tabulated in Appendix A.

To increase power to detect genetic association with ECC, GWAS associations for each variant available across all five cohorts were combined via meta-analysis. Summary statistics were meta-analyzed using Stouffer’s method in METAL [27]. Combined z-scores were calculated for each SNP with minor allele frequency (MAF) greater than 5%. A genomic control correction was not applied. To account for multiple testing, genome-wide and suggestive significance thresholds were set to *p*-value < 5 × 10^−8^ and <1 × 10^−5^, respectively. Manhattan and quantile–quantile (qq) plots were created in the qqman package in R (R Foundation for Statistical Computing, Vienna, Austria), and LocusZoom was used to visualize GWAS regions of interest [28].

### 2.4. Transcriptome-Wide Association Study (TWAS)

A TWAS was conducted to test for ECC association with predicted expression of gene transcripts. Expression was predicted based on summary statistics from the meta-analysis of ECC GWASs, integrated with common *cis* expression quantitative trait locus (eQTL) weights from multivariate adaptive shrinkage (MASHR) models across 49 tissues from the Genotype-Tissue Expression Project (GTEx, v8) and with linkage disequilibrium (LD) information from the 1000 Genomes Project Phase 3 reference panel in S-PrediXcan [29]. Since tissues known to be involved in the caries process (teeth, oral mucosa) were unavailable in GTEx, the expression weights from all 49 GTEx tissues were used. To increase power to detect transcript associations, the S-PrediXcan results in available tissues were meta-analyzed using S-MultiXcan [29].

Statistical significance was set taking into account multiple testing (Bonferroni correction, based on number of genes tested); the threshold for significance was set to *p*-value < 3.09 × 10^−6^, and the suggestive significance threshold was set to *p*-value < 1.0 × 10^−4^. Because TWAS often identifies multiple transcripts from a specific locus, associated transcripts were grouped by location of the corresponding gene locus if they were found within 1 Mb of one another. Manhattan and quantile–quantile (qq) plots were created in R. To reduce false positives due to LD misspecification (inaccurate TWAS results due to differences in LD between the GWAS study population and the eQTL reference panel), we flagged as suspicious any TWAS-identified transcripts that did not have a single transcript-tissue association that surpassed suggestive significance (*p*-value < 1.0 × 10^−4^) [30]. Details are available in the Appendix A.

Colocalization analysis was used to address the potential for spurious associations in TWAS results due to LD among variants used to predict gene expression. Here, the coloc R package was used to test for *cis*-QTL colocalization with GWAS signals. Program inputs were GTEx V8 eQTL data (gtexportal.org (accessed on 20 September 2021)) and GWAS summary statistics, limited to 1 Mb around the TWAS-identified genes. The posterior probabilities of five hypotheses were tested, H0: neither the GWAS-identified SNPs nor any eQTLs have associations in the region, H1: only SNPs show genetic association in the region, H2: only eQTLs show genetic association in the region, H3: both SNPs and eQTLs are associated but with different causal variants, and H4: SNPs and eQTLs are associated with the same causal variant. Evidence of SNP colocalization was defined by meeting three posterior probabilities (PP) based on the five hypotheses, as suggested by Barbeira et al.: (i) PP of H4 > 0.5, (ii) PP of H3 < 0.5, and (iii) PP of H0 + H1 + H2 < 0.3 [30]. The workflow for all analyses is summarized in Figure 1.

## 3. Results

### 3.1. GWASs and GWAS Meta-Analysis

The five GWASs yielded regions associated with ECC at suggestive significance (*p*-value < 5 × 10^−6^): COHRA1 (6 loci), COHRA2 (10), IFS (6), IHS (6), and ALSPAC (4) (Appendix A) and no regions at genome-wide significance. Select positional candidate genes within 500 kb of suggestive lead SNPs are annotated for putative relationships to ECC in Appendix A. Genes with functions related to tooth morphology, immune response to bacteria, nociception, periodontal disease, and other roles relevant to ECC were represented.

Using the results of the GWASs of ECC conducted on each of the five cohorts individually, we conducted a meta-analysis for 3,988,879 SNPs overlapping across the cohorts, totaling 2974 individuals (Table 1).

There were no variants associated with ECC at genome-wide significant levels. SNPs at seven unique loci surpassed suggestive significance (Figure 2, Table 2, Appendix A). The lead variants at seven loci had consistent directions of effect across cohorts except for rs9889096 in IHS. Notable positional candidate genes are annotated in Appendix A and discussed in the Appendix A.

### 3.2. TWAS

TWAS analysis showed that the genetically predicted expression of six genes, *LINC02905* (formerly *C8Orf49*), *CDH17*, *TAS2R43*, *SMIM10L1*, *TAS2R14*, and *NRAD1* (formerly *LINC00284*), was associated with ECC (*p*-value < 3.09 × 10^−6^). There was suggestive evidence for association (*p*-value < 1.0 × 10^−4^) with ECC for three additional transcripts: *TAS2R31*, *LACC1*, and *IGSF5.* Four of the nine transcripts identified in the TWAS did not reach the threshold for suggestive evidence of association (p_i_best < 10^−4^) in the individual gene–tissue analyses and thus, were flagged because they may represent false positives in the final TWAS due to LD misspecification: *LINC02905*, *NRAD1*, *LACC1*, and *IGSF5*. These were not further considered. Similarly to GWAS identifying multiple variants in linkage disequilibrium (LD) at one locus, TWAS often identifies multiple correlated transcripts per locus, many which may not be causal for the trait [31]. The nine transcripts were grouped into five loci; loci with multiple associated transcripts included: 12p13.2 (*TAS2R43*, *SMIM10L1*, *TAS2R14*, *TAS2R31*) and chr13q14.11 (*NRAD1*, *LACC1*). At 12p13.2, *TAS2R43* showed the strongest evidence of association based on *p*-value and effect size; however, *TAS2R14* has a direct relationship with cariogenic bacteria, discussed below. A Manhattan plot of TWAS results can be found in Figure 3 with the corresponding qq plot in Appendix A. TWAS-identified transcripts are listed in Table 3 with corresponding gene function annotations in Appendix A and full S-MultiXcan output in Appendix A.

Three of the transcripts identified encode members of the TAS2R bitter taste receptor family (*TAS2R43*, *TAS2R14*, and *TAS2R31*) [32]. Variants in *TAS2R43* that are related to greater perception of bitter taste (i.e., the functional version of the protein) correlate with liking coffee, and the locus containing *TAS2R43* explains ~9% of variation in perceived caffeine bitterness [33]. In this study, predicted increased *TAS2R43* expression positively correlated with ECC risk (z-score mean 1.80, SD +/−1.89). *TA2R43* and *TAS2R31* are also expressed in polymorphonuclear neutrophils, the earliest immune cells recruited to the site of inflammation [34]. TAS2R14 transduces quorum-sensing molecules from *S. mutans* to mediate the innate immune response in gingival epithelial cells [35]. 

The other transcripts identified in the TWAS did not have known biological relationships with caries. *CDH17* encodes cadherin 17, a peptide transporter in the small intestine and adhesion molecule that influences permeability of the intestinal epithelium [36]. *SMIM10L1* encodes small integral membrane protein 10 like 1, which is most highly expressed in adrenal glands in mice [37].

Colocalization of eQTLs and GWAS signals was performed at the nine TWAS-identified loci across all tissues where gene expression data were available. No loci showed evidence of LD-induced spurious association (PP of H3 < 0.5), no variants colocalized (PP of H4 > 0.5), and all loci showed limited power to detect colocalization (PP of H0 + H1 + H2 > 0.3) [30]. Thus, no TWAS-identified genes were screened out for potential LD contamination (Appendix A).

## 4. Discussion

We performed GWAS of ECC in five cohorts of European-ancestry children from various regions in the U.S. and England, performed a meta-analysis of overlapping SNPs, and prioritized genes via a cross-tissue TWAS, identifying four transcripts whose imputed expression is associated with ECC (*CDH17*, *TAS2R43*, *SMIM10L1*, *TAS2R14*). Our study is an important contribution toward our understanding of the genetic architecture of early childhood caries.

Notably, we identified a group of bitter taste receptor transcripts (*TAS2R43*, *TAS2R14*, *TAS2R31*) that have possible relevance to cariogenesis. The perception of bitter taste has long been known to influence caries risk, with bitter non-tasters more susceptible to caries [38]. Variants near taste receptor genes were previously associated with caries in adults, namely *TAS2R38, TAS2R3, TAS2R4, TASR25* [11]. A candidate gene study found that the taste receptor genes, *TAS2R38* (bitter) and *TAS1R2* (sweet), were associated with dental caries in primary and mixed dentition, respectively [39], in the COHRA1 cohort, one of the five cohorts included in this study.

Aside from their influence on taste preferences, taste receptors impact caries through mediation of oral host–microbial interaction [40]. The TAS2R bitter taste receptor family is involved in the perception of bitterness in taste buds [32], and its members are critical to host innate immune response to bacteria, including in periodontitis and to the cariogenic *S. mutans*. Specifically, TAS2R14 responds to a quorum-sensing molecule secreted by *S. mutans* and consequently induces a rapid innate immune response in gingival epithelial cells (Medapati et al. 2021). When stimulated by *S. aureus*, TAS2R14 mediates cellular defensin-β 2 secretion, an antimicrobial peptide implicated in caries, while *S. mutans* stimulation increases secretion of IL-8 [41]. *TAS2R43* and *TAS2R31* are known to be expressed in innate immune cells at sites of inflammation [34], supporting a potential role in oral host–microbial interaction. Similarly to TAS2R14, TAS2R38 has been found to transduce bacterial quorum-sensing molecules to stimulate the mucosal innate immune response in the upper airway [42] and gingival epithelial cells [43], while activation of the sweet receptor, TAS1R2, suppresses TAS2R-dependent antimicrobial peptide secretion in nasal cells and is likely deactivated in response to bacterial consumption of glucose in nasal secretions [44]. Sweet taste sensation diminishing antimicrobial secretions may also prove to be a relevant mechanism in the context of oral microbiome dysbiosis in caries.

Caries-related genetic factors, such as variation in bitter taste receptor genes, may be mediated by dietary factors. Genetic variants in taste receptor genes impact taste perception [45], and both bitter and sweet taste perception differ between children with and without caries [46,47]. Variants in *TAS2R43* are also associated with liking coffee, which is mediated by caffeine perception. Although American guidelines advise against caffeine intake for children under age 12, caffeine consumption is prevalent in the communities in West Virginia from which the COHRA1 and COHRA2 cohorts are partially drawn (*personal communication with Dr. Richard Crout*). In line with these observations, in some American communities, 15.2% of 2-year-olds have been found to consume up to 4 oz of coffee [48]. Coffee intake in children is associated with severe childhood obesity [49], possibly because of the sweeteners and creamer typically consumed with coffee drinks. Other commonly ingested caffeinated drinks, such as soft drinks, are cariogenic due to their acidic and highly sweetened nature [50]. It is possible that the association between genetic variation in caffeine perception and ECC is mediated by the cariogenic nature of caffeine-containing sweetened drinks since coffee in isolation is anticariogenic against *S. mutans* [51]. Further study of the genetics of taste perception in caries is warranted. Although the *CDH17* transcript identified using TWAS (*p*-value 3.32 × 10^−8^) does not have a direct connection with caries in the literature; it stands out as one of the most significant and reliable TWAS associations. CDH17 influences the permeability of the intestinal epithelium [36]. The integrity of the intestinal epithelium is increasingly understood to be important for prevention of systemic chronic inflammation, a state that predisposes to a wide variety of chronic diseases [52,53], some of which show overlap in heritability with dental caries [6].

It is important to note that the TWAS analyses cannot determine causation, and that the most significant transcript is not necessarily the causal one at loci with multiple associations, such as the locus harboring the bitter taste receptor genes [54]. Multiple transcripts at one locus can be identified by TWAS due to co-regulation; thus, the strength of the TWAS association should be a consideration when prioritizing follow-up studies of associated transcripts at one locus [31]. In the absence of definitive information about the causal tissues involved in caries, the S-MultiXcan TWAS approach analyzing all 49 GTEx tissues increased power to detect gene-ECC association, providing strong evidence for the *TAS2R* transcripts being associated with ECC risk. Based on these results, we nominate the previously identified *TAS2R38* and *TAS1R2*, along with *TAS2R43* and *TAS2R31* identified in this study, for future study to better characterize their potential roles in caries etiology. Furthermore, follow-up studies of TAS2R14 in ECC are supported by both biological evidence of its sensing of cariogenic bacteria and TWAS association results.

This study has a few limitations. The statistical power is limited by the smaller cohort size of the meta-analysis. Nevertheless, given how few studies there are of young children with detailed dental and genetic data, this represents the largest effort so far to identify genetic variants associated with ECC. There were also differences in the ECC phenotype definition between cohorts which may have impacted power to detect association—three cohorts did not include white spot classification and/or the missing teeth component as part of the ECC definition (IHS, IFS, ALSPAC). However, these limitations would very likely reduce the power to detect association but not be expected to result in spurious associations. Furthermore, a minority of children with ECC have missing teeth due to caries (for example, 11.7% of children with ECC in COHRA1 have missing teeth), and the increased heterogeneity in case definition between cohorts was outweighed by the benefits of increased power due to increased sample size. These power concerns were mitigated by placing greater weights on individual GWASs of the larger and more precisely phenotyped cohorts (COHRA1, COHRA2, ALSPAC) in the meta-analysis. We also accounted for potential sources of bias inherent to summary TWAS methods; however, the TWAS analysis is also under-powered to some degree. During the timeframe of data collection for the five studies (1990s–2010s), there were some changes in caries rates and diet, and these factors along with socioeconomic differences between sites could have contributed to increasing heterogeneity among individual study results and served to wash out potential signals in the meta-analysis and subsequent TWAS. This time period saw general decreases in caries prevalence in kids aged 2–5 aside from slight increases between 1988–1994 and 1999–2004 in the U.S. [55], while in England caries rates in the 1990s during ALSPAC sampling were comparable to caries rates in the U.S. in 2005 during COHRA1 data collection [1]. Excess sugar consumption in the U.S., a risk factor for caries, had increased from 1977 up to 2003 and generally decreased since then but remains above recommended levels [56,57]. Finally, due to the need to control for population stratification, solely European-ancestry individuals were included in this study. Thus, our findings, while being generalizable across socioeconomic classes and European-ancestry populations, are likely not directly generalizable to additional races and ethnicities. For additional discussion of findings, please see the Appendix A.

## 5. Conclusions

We report on the largest hypothesis-free genetic study of ECC to date, where we integrate GWAS data from five cohorts with existing gene expression datasets to identify four gene transcripts associated with the disease. We nominate for future study novel genetic risk factors for the condition with clear relevance to caries biology, including bitter taste receptors from the TAS2R family. Such genetic studies of caries are gradually improving our understanding of the process behind caries development and may help caries risk prediction in children to better target preventive or treatment measures.

## Figures and Tables

**Figure 1 genes-14-00059-f001:**
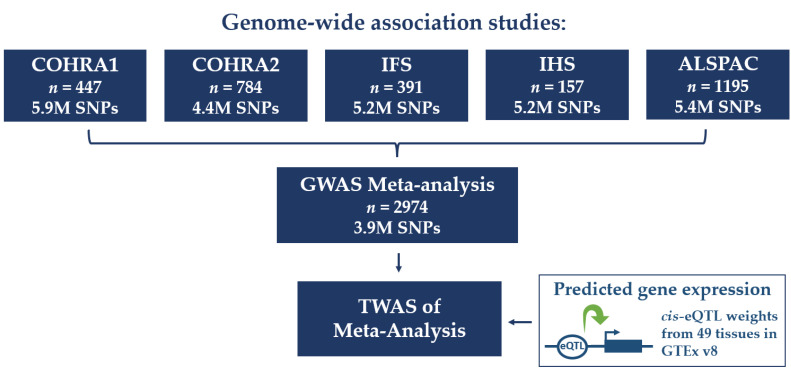
Flowchart of analyses performed. Genome-wide association studies (GWASs) of early childhood caries (ECC) were performed separately in five cohorts and meta-analyzed. The transcriptome-wide association study (TWAS) was performed using meta-analysis summary statistics integrated with predicted gene expression derived from common *cis* expression quantitative trait locus (eQTL) weights from multivariate adaptive shrinkage (MASHR) models across 49 tissues from Genotype-Tissue Expression (GTEx) project. TWAS analyses were performed in individual tissues in S-PrediXcan and meta-analyzed in S-MultiXcan.

**Figure 2 genes-14-00059-f002:**
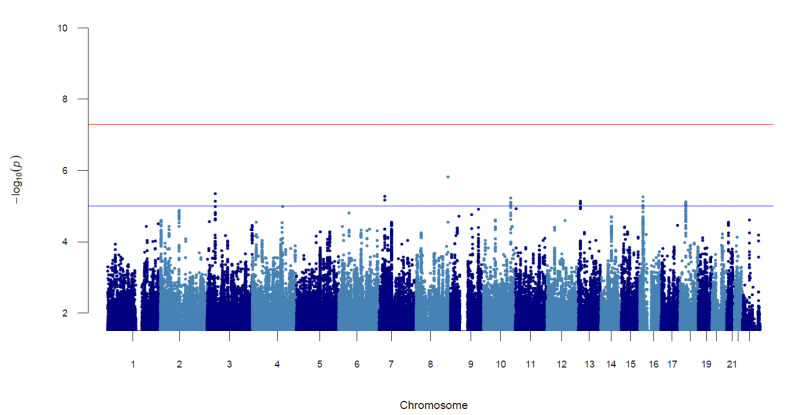
Manhattan plot of the meta-analysis results of the five early childhood caries (ECC) genome-wide association studies (GWASs). The horizontal red line represents the genome-wide significance threshold (*p* = 5 × 10^−8^), and the horizontal blue line is the suggestive significance threshold (*p* = 1 × 10^−5^). Each point represents a variant tested for association with ECC. The *x*-axis is the genomic position of the corresponding variant, and the *y*-axis is the negative logarithm of the association *p*-value. Seven loci surpassed the suggestive threshold of association with ECC, and none surpassed the genome-wide significance threshold.

**Figure 3 genes-14-00059-f003:**
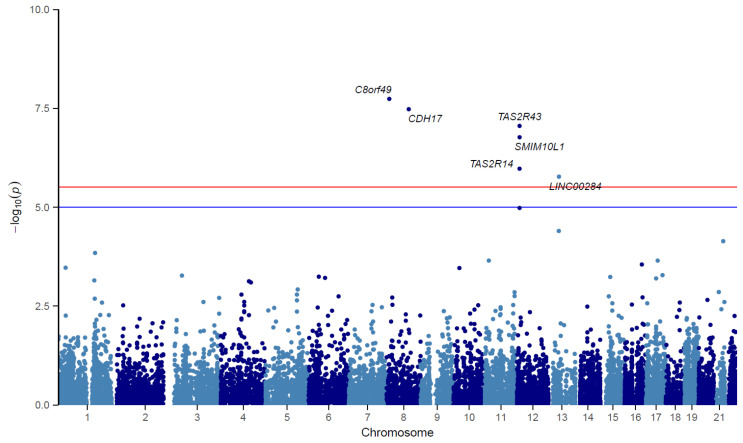
Manhattan plot of TWAS of ECC depicting transcripts associated with the condition. The horizontal red line represents the genome-wide significance threshold (*p* = 3.09 × 10^−6^), and the horizontal blue line represents the suggestive significance threshold (*p* = 1.00 × 10^−4^). Each point represents a transcript tested for association with expression imputed from the ECC GWAS meta-analysis. The *x*-axis is the genomic position of the gene corresponding to the transcript tested, and the *y*-axis is the negative logarithm of the ECC association *p*-value. The six transcripts surpassing significant *p*-value thresholds are annotated. *TAS2R43*, *SMIM10L1*, and *TAS2R14* are located at the same locus.

**Table 1 genes-14-00059-t001:** Characteristics of cohorts used in in genome-wide association studies (GWASs) of early childhood caries (ECC).

Cohort	Affected	Unaffected	Total	Age Mean (SD)	Female (%)	Location
COHRA1	176 (39.5%)	269 (60.5%)	447	3.88(1.37)	215 (48.1%)	PA, WV
COHRA2	156 (20%)	623 (80%)	784	3.80(1.52)	373 (47.9%)	PA, WV
Iowa Fluoride (IFS)	137 (35%)	254 (65%)	391	5.12 (0.37)	202 (51.7%)	IA
IOWA Head Start (IHS)	54 (34.4%)	103 (65.6%)	157	3.96(0.78)	76 (48.4%)	IA
ALSPAC	320 (26.8%)	875 (73.2%)	1195	4.83(0.77)	535 (44.8%)	SW England

PA Pennsylvania, USA; WV West Virginia, USA; IA Iowa, USA, SW; South West.

**Table 2 genes-14-00059-t002:** Lead single nucleotide polymorphisms (SNPs) suggestively associated (*p*-value < 1 × 10^−5^) with early childhood caries (ECC) in the meta-analysis.

SNP	Chr:Pos	Effect Allele/ Non-Effect Allele	MAF	Z score	Direction †	*p*	Select Gene(s) ‡
rs76823412	8:134690570	T/C	0.47	4.806	+++++	1.54 × 10^−6^	*ST3GAL1 **
rs74470773	3:33958083	T/C	0.47	−4.589	-----	4.46 × 10^−6^	*PDCD6IP **
rs1044956	7:24854765	A/G	0.49	4.551	+++++	5.35 × 10^−6^	*OSBPL3* *, *NPVF*
rs9889096	16:13575143	T/G	0.50	−4.541	--+--	5.59 × 10^−6^	*SHISA9 **
rs563135	10:115067899	A/C	0.51	−4.532	-----	5.86 × 10^−6^	*TCF7L2* *, *CASP7*
rs7325099	13:28104496	A/C	0.50	4.485	++?++	7.28 × 10^−6^	*LNX2* *, *POLR1D*, *RPL21*
rs8091366	18:24715618	A/G	0.52	4.476	+++++	7.60 × 10^−6^	*CHST9* *, *KCTD1*

Chr:Pos: chromosome and base pair position in GRCh37. MAF: minor allele frequency. Z score: combined Z statistic for the effect allele. †: order of cohorts represented, COHRA1, COHRA2, IHS, IFS, ALSPAC. +: positive direction of effect. -: negative direction of effect. ?: variant not examined. ‡: select genes within 500 kb of lead SNP. *: gene nearest associated SNP.

**Table 3 genes-14-00059-t003:** Genes corresponding to transcripts associated with early childhood caries (ECC) as identified by transcriptome-wide association study (TWAS). Genes encoding transcripts associated at significant (*p*-value < 3.09 x 10^−6^) or suggestive (*p*-value < 1.0 × 10^−4^) significance levels in S-MultiXcan analysis are shown. Genes are annotated for any relationship with early childhood caries.

Gene	Locus	TWAS *p*-Value	Gene Start Chr:Pos *	N Tissues †	Z Score (±SD)
***LINC02905*** (***C8Orf49***) ‡	chr8p23.1	1.82 × 10^−8^	8:11618765	9 (3)	−0.35 (1.33)
** *CDH17* **	chr8q22.1	3.32 × 10^−8^	8:95139394	23 (5)	1.74 (3.00)
** *TAS2R43* **	chr12p13.2	8.78 × 10^−8^	12:11243886	27 (3)	1.80 (1.89)
** *SMIM10L1* **	chr12p13.2	1.70 × 10^−7^	12:11323780	46 (3)	−0.35 (1.62)
** *TAS2R14* **	chr12p13.2	1.06 × 10^−6^	12:11090853	48 (4)	0.47 (1.27)
***NRAD1*** (***LINC00284***) ‡	chr13q14.11	1.69 × 10^−6^	13:44596471	30 (5)	0.13 (1.07)
*TAS2R31*	chr12p13.2	1.04 × 10^−5^	12:11182986	41 (4)	−0.25 (1.49)
*LACC1* ‡	chr13q14.11	3.98 × 10^−5^	13:44453420	48 (4)	0.10 (1.16)
*IGSF5* ‡	chr21q22.2	7.22 × 10^−5^	21:41117334	27 (11)	0.63 (1.40)

* Gene start position in chromosome:base pair format in GRCh37 coordinates. † N tissues: number of tissues where eQTL data were available for this gene; in parentheses are the N independent tissues—number of independent components of variation kept among the tissues’ predictions, i.e., synthetic independent tissues. SD: standard deviation. **Bolded** genes are associated with ECC at significant levels (*p* < 3.09 × 10^−6^); the rest at suggestive levels (*p* < 1.00 × 10^−4^) in the all-tissue TWAS. ‡: genes that do not surpass suggestive significance in any individual gene-tissue analysis (*p* < 1.0 × 10^−4^); these genes may be identified in all-tissue TWAS due to linkage disequilibrium (LD) misspecification.

## Data Availability

Underlying data used for the COHRA1, COHRA2, Iowa Fluoride, and Iowa Head Start studies are available via application to dbGAP. The data can be found at dbGAP accession numbers phs001591.v1.p1 and phs000095.v4.p2.

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
