# Peer review of "Association of Early Childhood Caries with Bitter Taste Receptors: A Meta-Analysis of Genome-Wide Association Studies and Transcriptome-Wide Association Study"

_genes, 2022, doi:10.3390/genes14010059_

Round 1

Reviewer 1 Report

Dear Authors,

Congratulations for this interesting research that includes extensive and up-to-date resources.

1.     What is the main question addressed by the research?

 The main subject of this study is a meta-analysis study aiming to identify genes with potential roles in ECC biology.

 2.     Do you consider the topic original or relevant in the field? Does it address a specific gap in the field?

It is an original and comprehensive meta-analysis study in the subject area. Recent studies in genetics examine the relationship between dental caries and taste receptors in detail.

 3.     What does it add to the subject area compared with other published material?

Reporting the largest hypothetical genetic study to date of ECC integrating GWAS data with existing gene expression datasets. They have nominated new genetic risk factors for future studies. It provides new information that may help predict caries risk in children.

 4.     What specific improvements should the authors consider regarding the methodology? What further controls should be considered?

The method is written in detail, but in a language that is difficult for readers to understand. It can be difficult to interpret metaanalysis studies, especially for readers unfamiliar with the genetic literature.

 5.     Are the conclusions consistent with the evidence and arguments presented and do they address the main question posed?

The results are consistent with the evidence presented and consistent with the studies reviewed. Notably, we identified a group of bitter taste receptor transcripts (TAS2R43, TAS2R14, TAS2R31) that have possible relevance to cariogenesis.

 6.     Are the references appropriate?

References are current and support the research.

 7.     Please include any additional comments on the tables and figures.

The tables are given clearly and precisely, but the figures may need a little more information.

Reviewer 2 Report

I have read the manuscript “Association of early childhood caries with bitter taste receptors:  a meta-analysis of genome-wide association studies and transcriptome-wide association study”, which is a large and interesting study that includes 5 cohorts in the analysis.

Overall this is an important study for the field that point some possible candidate genes for ECC.

I think it is necessary to rewrite the abstract. It is unclear that it is a GWASs of ECC in five cohorts, followed by a meta-analysis.

Although there is a reference, each cohort should be better described in the method, the age that ECC was evaluated should be clear in each cohort.

Were dietary factors information collected? Or taste preference evaluated? This should be highlighted at least in the discussion.

A flowchart would be useful.

The authors should discuss the variations in the characteristics of their cohorts. Including the collection period, and possible difference in caries prevalence and diet in this period.

A conclusion is necessary.
